# Targeting ABCC6 in Mesenchymal Stem Cells: Impairment of Mature Adipocyte Lipid Homeostasis

**DOI:** 10.3390/ijms23169218

**Published:** 2022-08-16

**Authors:** Ricarda Plümers, Michel R. Osterhage, Christopher Lindenkamp, Cornelius Knabbe, Doris Hendig

**Affiliations:** Herz- und Diabeteszentrum Nordrhein-Westfalen, Institut für Laboratoriums- und Transfusionsmedizin, Universitätsklinik der Ruhr-Universität Bochum, 32545 Bad Oeynhausen, Germany

**Keywords:** ABCC6, adipogenesis, lipid homeostasis, mesenchymal stem cells, metabolic disease, pseudoxanthoma elasticum

## Abstract

Mutations in ABCC6, an ATP-binding cassette transporter with a so far unknown substrate mainly expressed in the liver and kidney, cause pseudoxanthoma elasticum (PXE). Symptoms of PXE in patients originate from the calcification of elastic fibers in the skin, eye, and vessels. Previous studies suggested an involvement of ABCC6 in cholesterol and lipid homeostasis. The intention of this study was to examine the influence of ABCC6 deficiency during adipogenic differentiation of human bone marrow-derived stem cells (hMSCs). Induction of adipogenic differentiation goes along with significantly elevated *ABCC6* gene expression in mature adipocytes. We generated an *ABCC6*-deficient cell culture model using clustered regulatory interspaced short palindromic repeat Cas9 (CRISPR–Cas9) system to clarify the role of ABCC6 in lipid homeostasis. The lack of ABCC6 in hMSCs does not influence gene expression of differentiation markers in adipogenesis but results in a decreased triglyceride content in cell culture medium. Protein and gene expression analysis of mature *ABCC6*-deficient adipocytes showed diminished intra- and extra-cellular lipolysis, release of lipids, and fatty acid neogenesis. Therefore, our results demonstrate impaired lipid trafficking in adipocytes due to *ABCC6* deficiency, highlighting adipose tissue and peripheral lipid metabolism as a relevant target for uncovering systemic PXE pathogenesis.

## 1. Introduction

The ectopic calcification of connective tissue, especially elastic fibers, is the main characteristic of pseudoxanthoma elasticum (PXE, OMIM 264800), a rare genetic multisystem disorder [1,2]. Yellow papular lesions at flexural body sites followed by excessive wrinkling are often the earliest indication of PXE, usually occurring in childhood or adolescence [3,4]. Hyperpigmentation (Peau d’orange), fracturing (angioid streaks), and bleeding in the calcified Bruch’s membrane may finally lead to loss of central vision [5,6]. Cardiovascular manifestations, including hypertension, reduced peripheral pulse amplitude, *claudication intermittens*, and cardiomyopathy, arise from segmental constrictions of small vessels by mineral deposits (atherosclerosis) [7,8].

In 2000, mutations in the ATP-binding cassette transporter ABCC6 were found to cause PXE [9]. To date, approximately 400 mutations in the *ABCC6* gene have been described in PXE patients [10]. The causal relationship between ectopic calcification and ABCC6 remains elusive because ABCC6 is mainly expressed in the liver and kidney but only in small amounts in peripheral tissue, such as skin [11]. Based on this expression pattern, the metabolic hypothesis of a so far unknown substrate, transported by ABCC6 from hepatocytes into the bloodstream, whose absence evokes the symptoms of PXE observed, was established [12]. The lipid transporting properties of a variety of ABC transporters and metabolic studies suggest an involvement of ABCC6 in lipid homeostasis. Hosen et al. were able to demonstrate in silico that a variety of lipid compounds and bile secretions, inter alia, are putative substrates of ABCC6 [13]. The activity of HMG-CoA-reductase and proprotein convertase subtilisin/kexin type 9 protein expression, both being essential in cellular lipid homeostasis, were found to be elevated in fibroblasts from PXE patients [14,15].

Hepatocytes receive triglycerides (TG), fatty acids, and cholesterol absorbed in the gastrointestinal system through the bloodstream either in its free form or encased in chylomicrons [16]. The latter belongs to the group of lipoproteins assembled from apolipoproteins and phospholipids. Cholesterol and TG are embedded in very low-density lipoproteins (VLDL) and secreted into the bloodstream to provide peripheral tissue with energy [17,18]. Lipoprotein lipase (LPL) catalyzes the hydrolysis of TG out of VLDL and, thereby, supplies the basis for fat synthesis and storage, for example, in adipose tissue [19,20]. The remaining cholesterol-rich low-density lipoproteins (LDL) are resorbed into the liver or serve as cholesterol suppliers for tissues. Starving cells upregulate their LDL receptor (LDLR) expression to equalize their cholesterol demand, as seen for PXE fibroblasts [14]. Excessive cellular cholesterol is ejected through ABC transporter ABCA1 and, subsequently, returned from the periphery to the liver in high-density lipoproteins (HDL) [21].

The adipose TG lipase (ATGL; also known as palatin-like phospholipase domain-containing protein 2, PNPLA2) hydrolyzes TG within adipocytes as the rate-limiting step in lipolysis [22,23]. Moreover, the TG household is dependent on de novo synthesis involving several enzymes, one of them being the elongation of very long chain fatty acids protein 3 (ELOVL3) [24].

Due to the limited cultivation of mature adipocytes, the isolation and differentiation of human mesenchymal stem cells (hMSCs) into adipocytes is a widely used technique to uncover adipose tissue metabolic pathways. The hMSCs are pluripotent stem cells mostly isolated from bone marrow and able to differentiate into osteoblasts, chondrocytes, and adipocytes [25,26]. The gene expression of master regulator peroxisome proliferator-activated receptor γ (PPARG) is elevated during adipogenic differentiation and, consequently, induces the expression of adipocyte markers, such as perilipin 1 (PLIN1) and leptin (LEP) [27,28].

Our aim was to differentiate hMSCs into adipocytes and quantify *ABCC6* gene expression to shed light on the participation of ABCC6 in adipocyte metabolism. In the next step, we intended to establish a clustered regulatory interspaced short palindromic repeat (CRISPR–Cas9) system to deplete *ABCC6* in hMSCs, followed by the analysis of lipid homeostasis in mature adipocytes.

## 2. Results

### 2.1. Diminished ABCC6 Expression in Differentiated hMSCs via CRISPR–Cas9 Genome Editing

The basal gene expression of *ABCC6* in mesenchymal stem cells was found to be comparable with fibroblasts and low compared to HepG2 (Appendix A). Real-time quantitative PCR (RT-qPCR) was performed after the third induction cycle (day 12) and at the end of the maintenance phase (19 days) to evaluate *ABCC6* gene expression in hMSCs during adipogenic differentiation. While there was no difference in the *ABCC6* gene expression on day 12 of differentiation in induced hMSCs compared to the maintenance control, *ABCC6* mRNA expression was significantly induced by 12.95-fold (±2.06) in hMSCs cultured in induction medium on day 19 relative to maintenance control (Figure 1A).

The hMSCs were transfected with a ribonucleoprotein (RNP) complex composed of ABCC6-specific crisprRNA (crRNA) and Cas9 endonuclease to achieve *ABCC6* clustered regulatory interspaced short palindromic repeat (CRISPR) Cas9 based genome editing and thereby reducing its expression. Transfection efficiency of 93.7% (±1.6%) was determined via a count of ATTO-550 positive cells by fluorescence microscopy (Appendix A). Due to the passage-limited differentiation ability of hMSCs, the latter were further analyzed and differentiated in cell pools rather than single cell clones. Despite the high transfection efficiency, a remaining portion of wild-type and heterozygous hMSCs needs to be considered as part of the knock-out model, in the following defined as *ABCC6*-deficient hMSCs.

The tracing of the *ABCC6* gene expression in *ABCC6*-deficient hMSCs revealed no change between the differentiation-inducing and differentiation control cultivation after day 12, but a 3.23-fold (±0.67) increase in differentiated hMSCs at day 19 (Figure 1A). Compared to wild-type hMSCs, the *ABCC6* gene expression was diminished by 73.4% (±4.1%) in *ABCC6*-deficient hMSCs at day 19 of differentiation.

Sanger sequencing of the guideRNA (gRNA) binding site was performed for hMSC-68m and hMSC-66f separately to further confirm *ABCC6* genome editing in hMSCs. (Figure 1B). While wild-type sequences were matching, the reference sequence (NG_007558.3) base calling for *ABCC6*-deficient hMSCs was inaccurate, starting three bases upstream of protospacer adjacent motif (PAM) sequence, indicating successful genome editing in this region. Off-target effects were not assumed as no alterations in the region of interest genome sequence were observed for the three most likely off-targets glutamate ionotropic receptor NMDA Type Subunit 1, high mobility group nucleosome binding domain 5 and piezo type mechanosensitive ion channel component 2 (Appendix A).

A T7 assay was applied to clarify the mutational efficiency in hMSCs pools by mismatch cleavage (Figure 1C). The genomic *ABCC6* amplification product length is 554 base pairs, with 304 base pairs before the gRNA and 227 base pairs after the PAM sequence. Gel electrophoresis pictures an amplification product of more than 500 bp length, identified as a wild-type product, in all samples without the addition of T7 endonuclease. The same is true for wild-type amplificates treated with T7 endonuclease, while the incubation of amplificates derived from *ABCC6*-deficient hMSCs DNA isolates with T7 endonuclease results in the additional detection of fragments with around 331 and 242 base pairs, respectively, identified as genome editing fragments. The intensity of the wild-type band was set relative to the cumulated intensity of wild-type and genome editing fragment bands in the same sample. The quantification of the wild-type fragment corresponds to 29.8% in hMSC-68m, respectively, 23.3% in hMSC-66f. Therefore, a mutagenesis rate of more than 70% can be considered and goes along with the relative *ABCC6* mRNA expression in these cell lines.

A TA cloning assay displayed several single allele mutations, all of them lacking either thymine or cytosine four, respectively, three bases upstream of the PAM sequence, indicating cleavage by Cas9 endonuclease in this region. The resulting mutations include deletion between 1 and 250 base pairs or insertions between 1 and 20 base pairs. Consequently, the wild-type 1503 amino acid sequence is truncated either by a few amino acids (in frame mutation) to less than 600 amino acids by a premature stop codon (out of frame mutation) or the intron-exon junction is depleted, resulting in the deletion of exon 12 (Appendix A).

### 2.2. Successful Differentiation of hMSCs into Adipocytes Is Largely Independent of ABCC6

Adipogenic differentiation of hMSCs was confirmed by the relative quantification of adipogenic differentiation markers *PPARG*, *LEP*, and *PLIN1* gene expression, as well as immunofluorescent staining for PLIN1 and co-staining for lipids in lipid droplets with bordipyrromethene (Bodipy).

The *PPARG* gene expression was significantly increased by 11.20-fold (±1.14) on day 12 and 10.18-fold (±2.06) at day 19 in differentiation-induced wild-type hMSCs compared to the differentiation control. Its gene expression in *ABCC6*-deficient hMSCs upon the administration of induction medium was significantly elevated 10.61-fold (±0.92) at day 12 and 7.97-fold (±1.07) at day 19. There was no notable difference in *PPARG* gene expression between wild-type and *ABCC6*-deficient hMSCs at either time point (Figure 2A).

The *LEP* gene expression at day 12 was not induced in either wild-type or *ABCC6*-deficient hMSCs when cultured in differentiation medium. Its gene expression at day 19 was significantly increased by 12.71-fold (±2.85) in wild-type and by 13.27-fold (±3.06) in *ABCC6*-deficient hMSCs compared to controls cultured in maintenance medium without a difference considering the two genetic backgrounds (Figure 2B).

The *PLIN1* mRNA expression was highly induced when culturing wild-type (day 12: 4086-fold ± 825.74; day 19: 9300.68-fold ± 1655.20) and *ABCC6*-deficient hMSCs (day 12: 5404.17-fold ± 1629.74: day 19: 4269.14-fold ± 904.77) in induction medium at both time points. Significant differences between induced wild-type and *ABCC6*-deficient hMSCs were not observed at day 12 but were at day 19, as *PLIN1* gene expression was reduced in *ABCC6*-deficient hMSCs by 37% (±6%) (Figure 2C).

Upon the cyclic administration of induction medium, lipids visualized by Bodipy staining and enclosed by PLIN1 were excessively stored in wild-type and *ABCC6*-deficient hMSCs compared to controls cultured only in maintenance medium (Figure 2D). Differences in the lipid droplet size or number were not detected between wild-type and *ABCC6*-deficient hMSCs (Appendix A). The mean grey value quantification of PLIN1 immunofluorescent labeling revealed a significantly higher PLIN1 load in *ABCC6*-deficient hMSCs of 40.8% (±11.0%) in hMSC-68m and 136.8% (±31.2%) in hMSCs-66f (Figure 2E).

### 2.3. ABCC6 Deficiency Impairs Lipid Homeostasis in Adipocytes

The TG content in cell culture supernatants of wild-type and *ABCC6*-deficient hMSCs was quantified to evaluated lipid homeostasis.

The TG content in the cell culture supernatant was significantly higher for all adipogenic-differentiated hMSCs compared to undifferentiated controls (hMSCs-68mwild-type: 3.30-fold ± 0.37, hMSCs-68m *ABCFC6*-deficient: 2.44-fold ± 0.21, hMSCs-66fwild-type: 3.85-fold ± 0.36, hMSCs-66f *ABCC6*-deficient: 2.97-fold ± 0.26) supporting the assumption of successful adipogenic differentiation (Figure 3).

The comparison of the TG content of cell culture supernatants reveals a significant reduction by 20% (±6%) for hMSC-68m and 19% (±6%) for hMSC-66f in *ABCC6*-deficient adipocytes relative to the wild-type control.

We performed mRNA gene expression analysis and immunofluorescent staining to further characterize the lipid homeostasis of adipocytes derived from hMSCs differentiated for 19 days. The lipid uptake was displayed by the expression pattern of *LDLR* and lipase *LPL*. Intracellular fatty acid trafficking and processing were evaluated exemplarily by *ATGL* and *ELOVL3* gene expression. Cholesterol efflux was assessed via *ABCA1* mRNA expression analysis.

The *LDLR* gene expression was significantly elevated by 2.36-fold (±0.27) in wild-type and by 1.49-fold (±0.18) in *ABCC6*-deficient adipocytes compared to undifferentiated hMSCs. The ABCC6-deficiency in differentiated adipocytes resulted in a significant 37% (±8%) reduction of *LDLR* gene expression relative to wild-type (Figure 4A).

The hMSCs under non-differentiating conditions were found not to express *LPL*, whereas expression was detected in mature adipocytes. The comparison of differentiated adipocytes reveals a significantly lower *LPL* gene expression upon *ABCC6*-deficiency (34% ± 8%) (Figure 4B).

The *ATGL* mRNA expression was induced significantly in wild-type by 10.19-fold (±0.89) and *ABCC6*-deficient adipocytes by 8.79-fold (±0.76) compared to their undifferentiated controls. Directly comparing its expression in adipocytes revealed a reduction by 17% (±6%) upon *ABCC6* deficiency (Figure 4C). Significantly reduced ATGL expression was confirmed by the immunofluorescent staining of ATGL in adipocytes at day 19 of differentiation (Figure 4D). The quantification of cellular mean grey value revealed a reduction of ATGL protein expression in ABCC6-deficient adipocytes by 31.0% (±5.4%) in hMSCS-68m and 52.3% (±3.9%) in hMSCS-66f (Figure 4E).

The induction of adipogenic differentiation resulted in a significantly elevated *ELOVL3* gene expression in adipocytes compared to the differentiation control (wild-type: 3.06-fold ± 0.32, *ABCC6*-deficient: 1.69-fold ± 0.19). The *ELOVL3* gene expression was 28% (±7%) lower in *ABCC6*-deficient adipocytes than in wild-type adipocytes (Figure 4F).

The *ABCA1* mRNA expression was found to be induced upon adipogenic differentiation in wild-type (2.13-fold ± 0.35) and *ABCC6*-deficient hMSCs (1.53-fold ± 0.22), while its expression was diminished by 24% (±7%) in *ABCC6*-deficient adipocytes when compared with the wild-type control (Figure 4G).

## 3. Discussion

The elucidation of the pathomechanism behind ABCC6 causing PXE is a critical step in providing adequate therapy to patients. The application of statins, such as atorvastatin, is currently state-of-the-art medication due to the high risk of coronary artery disease and altered serum lipid profile in patients [29,30]. Positive effects of atorvastatin on abnormal cholesterol biosynthesis and partial improvement of the inflammatory phenotype in the periphery represented by PXE fibroblasts underline the systemic impact of ABCC6 [14,31].

The possibility of ABCC6 being involved in lipid metabolism is further supported by studies in polar bears. Among other genes, *ABCC6* is under positive selection compared to brown bears and, thereby, might be potentially associated with adaption to cold temperatures [32]. Moreover, *ABCC6* is one of 73 genes whose gene expression in adipose tissue is significantly decreased in obese patients responding to caloric restriction compared to low responders [33]. Bariatric surgery results in the downregulation of *ABCC6* mRNA expression in subcutaneous adipose tissue as well [34]. In silico screening mostly revealed lipid compounds and biliary secretions as potential substrates of ABCC6 [13]. Taken together, these studies imply that ABCC6 is a potent regulative element in lipid metabolism.

In this study, our aim was to support the hypothesis of ABCC6 influencing lipid metabolism and provide insight into the underlying mechanism in adipocytes.

For the first time, we were able to describe a correlation between the *ABCC6* mRNA gene expression and the differentiation of hMSCs into adipocytes. Thereby, we were able to underline the relevance of ABCC6 in adipose tissue. In order to further clarify the involvement of ABCC6 in adipocytes, CRISPR–Cas mediated knock-out was established. Since the differentiation ability of hMSCs is significantly reduced with repeated subculturing, clonal expansion had to be omitted in this model. A remaining fraction of wild-type or heterozygous cells in the population had to be accepted and needs to be considered when classifying the present results. Nevertheless, the influence of the expressed wild-type ABCC6 can be regarded as low since the mutational rate determined via a T7 assay is estimated to be more than 70%, and there is a significantly reduced *ABCC6* mRNA expression in cells regarded as differentiated ABCC6-deficient adipocytes. The PPARγ is the main regulator of adipogenic differentiation, and its expression is an adequate indicator to evaluate the influence of genetic deletions on adipogenesis. In the case of *PPARG* knock-out, the formation of adipose tissue in mice is restricted, and PPARg inhibition by berberine diminishes the storage of lipid in 3T3-L1 preadipocytes [35,36]. We found adipose differentiation marker *PPARG* and *LEP* gene expression in adipocytes greatly increased upon the administration of induction medium. Moreover, the formation of lipid droplets was observed only in adipogenic-induced cells. These results confirmed a successful induction of adipogenic differentiation. Consequently, hMSCs can be considered mature adipocytes after 19 days of culture in differentiation conditions. A direct influence of ABCC6 on adipocyte differentiation in our model is not assumed because the *PPARG* gene expression and lipid droplet number and size were independent of *ABCC6* genome editing together with the gene expression pattern of *ABCC6* during differentiation. This leads to the hypothesis of ABCC6 being part of lipid homeostasis in mature adipocytes. mRNA gene expression analyses, as well as immunofluorescence-based protein detections, were performed to gain a first impression of the extent to which this hypothesis is reflected in the generated ABCC6-deficient adipocytes.

The gene expression of *PLIN1*, a target gene of PPARg [37], was reduced in *ABCC6*-deficient mature adipocytes compared to the wild-type control. Apart from that, immunofluorescent staining of PLIN1 revealed a higher load in *ABCC6*-deficient adipocytes. Perilipins are lipid droplet surface proteins stabilizing and protecting them from lipase action by binding 1-acylglycerol-3-phosphate O-acyltransferase [38]. The latter is a co-activator of ATGL released upon PLIN1 phosphorylation and, thereby, allowing lipolysis. The perilipin load is regulated transcriptionally, mainly by PPARg, and posttranslationally by proteasomal degradation after lipolysis [39,40]. The discrepancy between the *PLIN1* gene expression pattern and PLIN1 protein load in this study may be due to missing lipolysis in the form of lipid droplet degradation and thereby accumulation of PLIN1, resulting in a feedback reduction of its gene expression.

This hypothesis of disturbed lipolysis goes along with the reduced *ATGL* gene expression in *ABCC6*-deficient adipocytes found in this study. The ATGL predominantly hydrolyzes triacylglycerol to diacylglycerol releasing free fatty acids [23]. An ATGL deficient mice model revealed a systemic impact on TG metabolism via accumulation in fat tissue and the subsequent elevation of hepatic fatty acid metabolism [41,42]. Our data present diminished ATGL expression in *ABCC6*-deficient adipocytes on a gene and protein expressional level. Consequently, a similar impact on the systemic TG household and hepatic metabolism, as observed in ATGL knock-out mice, can be guessed and might be further confirmed by data from an *Abcc6*^−/−^ mouse model or in vitro co-cultivation of *ABCC6*-deficient adipocytes and hepatocytes.

The TG content was quantifiable in our study and significantly reduced in the *ABCC6*-deficient culture of mature adipocytes, supporting the hypothesis of lipid accumulation and impaired lipolysis in adipocytes. Brampton et al. found the TG concentration in an Abcc6-deficient mouse model to be elevated [43]. The chance that a diminished release of TG from adipose tissue may result in altered systemic effects, for example, elevated hepatic fatty acid metabolism or altered uptake from food as described for ATGL, should be considered [41,42]. Our model helps to put the focus on adipose tissue in this regulatory network, mirroring the potential systemic effect of impaired fatty acid release from adipocytes.

Furthermore, TG metabolism is affected by the extra-cellular hydrolysis of TG from VLDL mediated by LPL. Upon fatty acid release, VLDL turns into LDL, being opsonized when bound to its receptor LDLR [44]. Both *LPL* and *LDLR* were found to be less expressed at the transcriptional level in *ABCC6*-deficient adipocytes speaking for a non-starving lipid status. Interestingly, the opposite has been observed in PXE fibroblasts underlining the systemic differences and a potential lipid overload in adipose tissue with a simultaneous shortage at symptomatic sites, such as the skin [14]. Although the LDL level in the cell culture supernatant was too low to be detected in this study, a systemic rise of LDL in the serum of PXE patients due to a missing uptake in the periphery can be supposed based on these results. An elevated LDL serum concentration contributes to atherosclerosis and, thereby, coronary artery disease, both of which are common characteristics of PXE disease [8,45]. Unfortunately, the LDL serum concentration in PXE patients is hard to evaluate as many patients are often treated with lipid-lowering drugs [46]. However, the double knock-out of *Ldlr* and *Abcc6* in mice not only leads to elevated LDL concentration in the serum but results in atherosclerosis [43]. As lipid metabolism differs significantly between mice and humans [47], the present study serves as a supplementary model for the investigation of lipid metabolism under *ABCC6*-deficient conditions.

Additionally, not only lipolysis but fatty acid neogenesis is potentially influenced by *ABCC6* deficiency, as indicated by the mRNA expression of *ELOVL3*. It was found to be decreased in *ABCC6*-deficient adipocytes. Little is known about the regulation of ELOVL expression, but it is assumed that it is mostly influenced at a transcriptional level by transcription factors, such as PPARγ, and cellular stimuli, such as polyunsaturated fatty acids [48]. Kobayashi and Fujimori observed a diminished lipid droplet accumulation in 3T3-L1 preadipocytes upon the administration of *ELOVL3* targeting siRNA [24]. As this has not been observed in our model, it is more likely that the *ELOVL3* gene expression is down-regulated because of intracellular fatty acid overload, possibly due to a missing secretion of lipid metabolites.

In addition to impaired lipolysis, fatty acid neogenesis, and uptake, the present study gives hints for diminished cholesterol efflux in *ABCC6*-deficient adipocytes, another building block in intracellular lipid trafficking. This is reflected by decreased *ABCA1* expression levels. ABCA1 is essential for the liberation of excess cholesterol from peripheral tissue in order to form HDL particles for reverse cholesterol transport to the liver [21,49]. Consequently, diminished cholesterol efflux from adipose tissue may result in reduced HDL serum levels, as has already been observed in *Abcc6*-deficient mouse models [43,50].

Our results correlate diminished intra- and extra-cellular lipolysis, fatty acid neogenesis, and the release of lipids in mature adipocytes with the genetic depletion of ABCC6. Reduced lipid trafficking might be the result of a missing lipid influx or efflux in adipocytes when ABCC6, known to be a transporter basolaterally expressed in hepatocytes, is depleted [51]. Further investigation of the transcriptome and metabolome in *ABCC6*-deficient adipocytes will be necessary to support our primary results within the context of common markers of lipid homeostasis to reveal the underlying pathomechanism.

For the first time, we were able to elucidate the influence of *ABCC6* on the lipid homeostasis of adipose tissue and, thereby, reveal another potential regulatory element in the complex PXE pathogenesis. The next step will be to bring these results into a systemic context uncovering the link between impaired lipid homeostasis due to ABCC6 deficiency and peripheral symptoms of PXE patients.

## 4. Materials and Methods

### 4.1. Cell Culture and Adipogenic Differentiation

Bone marrow-derived hMSCs were purchased from Promocell (Heidelberg, Germany). Information about their characteristics is listed in Table 1. Cells were cultivated in mesenchymal stem cell basal medium™ (MSCGM, Lonza, Basel, Switzerland) supplemented with hMSC SingleQuot™ (Lonza, Basel, Switzerland) according to the manufacturer’s instructions. The hMSCs were subcultured upon 90% confluence.

Regarding the comparative *ABCC6* gene expression analysis, NHDF (*n* = 2; obtained from the Coriell Institute for Medical Research, Camden, NJ, USA) and HepG2 (American Type Culture Collection, Manassas, VA, USA) were cultured in Dulbecco’s Modified Eagle’s Medium supplemented with 10% *v*/*v* fetal calf serum (Gibco, Waltham, MA, USA), 4 mM L-glutamine (PAN Biotech, Aidenbach, Germany), and 1% (*v*/*v*) penicillin-streptomycin-Amphotericin B solution (100×; PAN Biotech, Aidenbach, Germany).

An hMSC Adipogenic Differentiation Medium BulletKit™ (Lonza, Basel, Switzerland) was utilized to attain adipogenic differentiation. In brief, adipogenic maintenance and induction media were prepared as stated in the manufacturer’s instructions. The hMSCs were seeded at a density of 21 × 10^4^ cells per cm^2^ and cultured until confluence was reached, typically within 7 days. Thereupon, three cycles of induction and maintenance were accomplished, each consisting of three days of culture in induction medium followed by one day of maintenance in media cultivation. The differentiation was finished after another week of cultivation in maintenance medium. The differentiation control was obtained by culturing hMSCs in parallel only in maintenance medium.

### 4.2. CRISPR–Cas9 Mediated ABCC6 Knock-Out

Knock-out of ABCC6 was achieved by application of an RNP complex formed by Cas9 Nuclease, a trans-activated crisprRNA (tracrRNA), and a guideRNA. ATTO™ 550 CRISPR–Cas9 tracrRNA, Alt-R S.p. Cas9 Nuclease V3, as well as predesigned Alt-R CRISPR–Cas9 guideRNA (Hs.Cas9.ABCC6.1.AC) were obtained from IDT (Coralville, IA, USA).

Alt-R CRISPR–Cas9 gRNA and ATTO™ 550 CRISPR–Cas9 tracrRNA were diluted to a 1 µM final concentration in nuclease-free IDTE buffer and incubated for 2 min at 95 °C to generate crRNA complex. The RNP complex formation was completed by mixing 11.5 µL of 1 µM crRNA and 23 µL of a 1 µM Alt-R S.p. Cas9 Nuclease V3 in 465.5 µL Opti-MEM™ (Life Technologies, Carlsbad, CA, USA) and keeping at room temperature for 5 min. Finally, 24.3 µL Lipofectamine™ CRISPRMAX™ (Thermo Fisher Scientific, Waltham, MA, USA) in a total volume of 500 µL Opti-MEM™ (Gibco, Waltham, MA, USA) was merged with the RNP complex solution. Following an incubation period of 20 min in the dark at room temperature, the mixture was brought into a 6-well plate (Greiner, Frickenhausen, Germany) cavity. A quantity of 3.6 × 10^4^ hMSCs resuspended in 2 mL antibiotic-free MSCGM™ was dropped onto the transfection mixture. Media was changed after 24 h and thereafter every 2 to 3 days.

Upon 90% confluence, cells were seeded for adipogenic differentiation.

### 4.3. Nucleic Acid Isolation

Isolation of DNA, RNA, and protein was performed using AllPrep DNA/RNA Mini Kit (Qiagen, Hilden, Germany) according to the manufacturer’s instructions. NanoDrop 2000 spectrophotometer (Peqlab, Erlangen, Germany) was used to measure nucleic acid concentrations.

### 4.4. Gene Expression Analysis

An amount of 1 µg total RNA was transcribed using SuperScript II Reverse Transcriptase (Thermo Fisher Scientific, Waltham, MA, USA) for complementary DNA (cDNA) synthesis. The qRT-PCR measurements were performed with reaction mixtures containing 5 µL LightCycler 480 SYBR Green I Master reaction mixture (Roche, Basel, Switzerland), 0.25 µL 25 µM forward and reverse primer (Biomers, Ulm, Germany), 2 µL water, and 2.5 µL 1:10 diluted cDNA. Primer sequences are listed in Table 2. Amplification and detection were accomplished on a LightCycler 480 Instrument II system (Roche, Basel, Switzerland). The PCR program included initial incubation for 5 min at 95 °C, 45 cycles of denaturation (95 °C, 10 s), annealing (specific annealing temperature, 15 s), and elongation (72 °C, 20 s). Following amplification, a melting curve analysis was performed. Relative mRNA gene expression was normalized on the relative succinate dehydrogenase complex flavoprotein subunit A (*SDHA*), hydroxymethylbilane synthase (*HMBS*), and ribosomal protein L13 (*RPL13*) gene expression was calculated based on the delta-delta Ct method considering PCR efficiency and internal calibration. Each condition was measured in biological and technical triplicates.

### 4.5. Genomic DNA Amplification

A HotStarTaq DNA polymerase kit was obtained from Qiagen (Hilden, Germany). The DNA template was diluted to a final concentration between 10 and 50 ng/µL in water. A volume of 29.3 µL water, 5 µL PCR buffer, 10 µL Q-Solution, 0.5 µL dNTPs (10 mM each), 1 µL 25 µM forward and reverse primers (Biomers, Ulm, Germany), 0.2 µL HotStar Taq polymerase, and 2.5 µL DNA template were mixed together. Primer sequences are listed in Table 3. Following an initial denaturation for 15 min at 95 °C, 35 cycles of denaturation (1 min, 95 °C), annealing (1 min, primer pair-specific annealing temperature), and elongation (1 min 72 °C) were performed. Final elongation was performed for 10 min at 72 °C.

The PCR products were purified using the MSB^®^ Spin PCRapace kit (Invitek Molecular, Berlin, Germany), according to the manufacturer’s instructions.

### 4.6. T7 Endonuclease I Mutagenesis Assay

Base pair mismatches were cleaved by T7 endonuclease I, which was therefore used to evaluate the genome editing efficiency.

A T7 endonuclease I kit was purchased from NEB (Ipswich, MA, USA). A combination of 200 ng genomic DNA amplificate and 2 µL 10× reaction buffer was heated to 95 °C and gradually cooled at a rate of −0.3 °C per s till 25 °C to form heteroduplexes. The subsequent addition of 1 µL endonuclease-I (10 U/µL) and incubation for 1 h at 37 °C led to mismatch cleavage. A volume of 1.5 µL of a 0.25 M EDTA solution was applied to stop the reaction. The DNA fragments were analyzed via electrophoresis in a 1.8% agarose gel. A pUC19 DNA/MspI (HpaII) Marker (Thermo Fisher Scientific, Waltham, MA, USA) was used as a standard. The band intensity was quantified using ImageJ 1.8.0 (National Institutes of Health, Bethesda, MD, USA).

### 4.7. TA cloning for Single Allele Sequencing

A genomic DNA amplificate was cloned into a pCR™ 2.1 Vector using the TA cloning kit (Invitrogen, Waltham, MA, USA). In brief, 1.5 µL of fresh amplificate, 2 µL 5× T4 DNA Ligase Reaction Buffer, and 2 µL of pCR™ 2.1 Vector were filled to a total volume of 9 µL with water. Finally, 1 µL of ExpressLinkTM T4 FNA Ligase was added. Ligation was performed overnight at 4 °C. For transfection, 3 µL of ligated vector were pipetted into 50 µL suspension of competent TOP10 *Escherichia coli*. Following 30 min incubation on ice, heat shock was performed by heating the bacteria for 30 s at 42 °C and subsequent cooling on ice for 2 min. After the addition of 500 µL S.O.C. medium (Thermo Fisher Scientific, Waltham, MA, USA) and shaking incubation for 1 h at 37 °C, bacteria were plated on lysogeny broth (LB) agar plates containing 75 µg/mL ampicillin. Single clone colonies were picked 24 h later and incubated for another 24 h in 5 mL LB liquid medium containing 75 µg/mL ampicillin. Plasmid isolation was attained using the QIAprep Spin Miniprep Kit (Qiagen, Hilden, Germany).

### 4.8. Sanger Sequencing

The sequencing reaction was prepared with a BigDye™ Terminator v3.1 Cycler Sequencing Kit (Thermo Fisher Scientific, Waltham, MA, USA). Either 2.5 µL of genomic DNA amplificate or 200–300 ng plasmid, 2 mL Premix, 1.5 µL 2.5 µM forward primer, and 4 µL BigDye buffer (5×) were filled up with water to a total volume of 20 µL for each sequencing reaction. Initial denaturation for 2 min at 95 °C was followed by 30 cycles of denaturation (10 s, 95 °C), annealing (10 s, primer-specific annealing temperature), and elongation (4 min, 60 °C). Reaction mixture purification was achieved by 4 min centrifugation at 1000× *g* on a 750 µL Sephadex™ G-50 column (Sigma-Aldrich, St. Louis, MO, USA). Sanger Sequencing was performed on an ABI Prism 3500 Genetic Analyzer (Thermo Fisher Scientific, Waltham, MA, USA).

### 4.9. Immunofluorescent Staining, Bodipy Staining, and Fluorescence Microscopy

Cells were seeded onto coverslips (ø 13 mm) for the fluorescence microscopy experiment. Following the cultivation procedure intended and washing twice in phosphate-buffered saline (PBS 1x; Gibco, Waltham, MA, USA), cells were fixed during 20 min incubation with 4% paraformaldehyde (Roth, Karlsruhe, Germany) and washed twice again with PBS. Procedures were executed at room temperature. Permeabilization was accomplished by 10 min incubation with 0.1% Triton^®^ X 100 in PBS and another two washing steps.

Bodipy staining for fatty acid labeling was performed by incubation with 5 µM Bodipy in PBS for 1 h, followed by two washing steps.

Immunofluorescent staining was accomplished beginning with an application of 5% bovine serum albumin (BSA) in PBS to block unspecific binding sites. Subsequently, cells were washed twice again before 2 h incubation with primary anti-PLIN1 antibody (ab3526; Abcam, Cambridge, UK) or anti-ATGL-antibody (ab207799; Abcam, Cambridge, UK) diluted 1:400 in 1% BSA in PBS. After another two washes, cells were incubated for 1 h in the presence of an Alexa Fluor 555-conjugated goat anti-rabbit antibody (ab150078; Abcam, Cambridge, UK) diluted 1:1600 in PBS. After double washing, cells were counterstained with 0.25 µM DAPI solution (Abcam, Cambridge, UK). Coverslips were again washed twice prior to mounting in ROTI^®^Mount FluorCare mounting media (Roth, Karlsruhe, Germany). Fluorescence images were captured using a BZ-X810 microscope (Keyence, Osaka, Japan). The mean grey value was determined using Image J (National Institutes of Health, Bethesda, MD, USA).

### 4.10. Triglyceride Quantification

Measurements of TG in cell culture supernatants were conducted using Alinity ci 2 (Abbott, Chicago, IL, USA). Results were calibrated on an internal standard (fetal calf serum, FCS; Gibco, Waltham, MA, USA) and decreased by the basal TG content in the cultivation medium.

### 4.11. Statistical Analysis

The data represented data are shown as mean values ± standard error of the mean. Statistical calculation was performed using GraphPad Prism 9 (GraphPad, San Diego, CA, USA), applying non-parametric two-tailed Mann–Whitney U tests. Statistical significance was assumed for probability (*p*) values equal to or below 0.05.

## Figures and Tables

**Figure 1 ijms-23-09218-f001:**
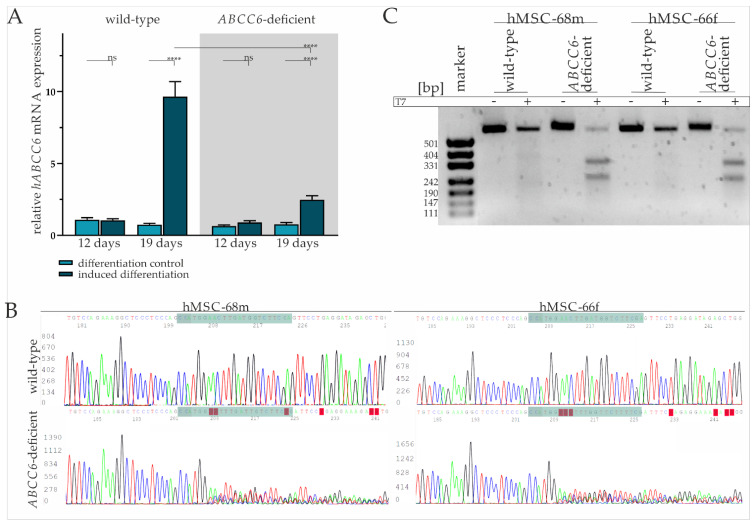
*ABCC6* gene expression during adipogenic differentiation and CRISPR–Cas9 mediated knock-out in hMSCs. (**A**) *ABCC6* mRNA expression in hMSCs during adipogenic differentiation was measured via RT-qPCR. hMSCs (*n* = 2) were cultured for differentiation over a period of 19 days, with 12 days marking the end of cyclic administration of adipogenic induction medium. *ABCC6*-deficient hMSCs (grey highlighted) were generated using the CRISPR–Cas9 system. Data are shown as mean ± standard error of the mean (SEM). Mann–Whitney U test significance levels: not significant (ns), *p* < 0.0001 (****). (**B**) Sanger sequencing of *ABCC6* exon 12 in the guideRNA binding region (green) in wild-type and *ABCC6*-deficient hMSC-68m and hMSC-66f. (**C**) Agarose gel electrophoresis of *ABCC6* exon 12 amplificates derived from wild-type or *ABCC6*-deficient hMSC-68m and hMSC-68f with or without additional incubation with T7 endonuclease.

**Figure 2 ijms-23-09218-f002:**
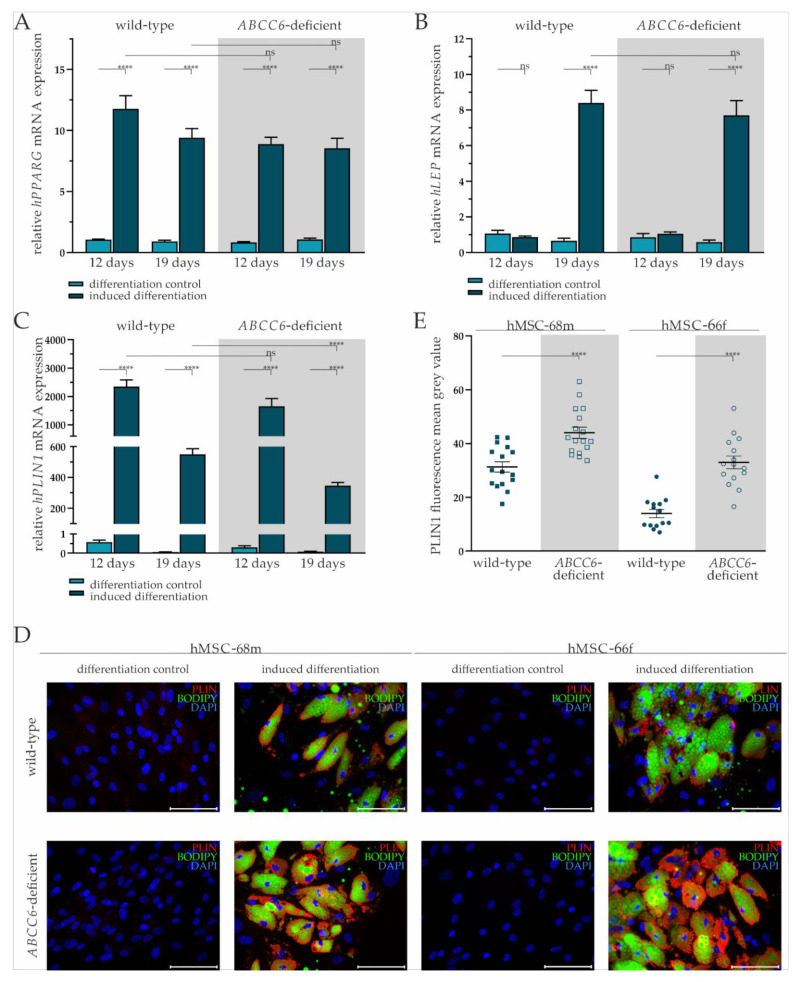
Gene expression of *PPARG*, *LEP*, and *PLIN* and fluorescent staining of PLIN1 and lipids in adipogenic differentiated hMSCs. (**A**) mRNA Expression of *PPARG*, (**B**) *LEP*, and (**C**) *PLIN1* in hMSCs during adipogenic differentiation was measured via RT-qPCR. Wild-type and *ABCC6*-deficient hMSCs (*n* = 2, grey highlighted) were cultured for differentiation over a period of 19 days, with 12 days marking the end of the cyclic administration of the adipogenic induction medium. (**D**) Immunofluorescent staining for PLIN1 (red), lipid staining via Bodipy (green), and nuclear counterstaining with DAPI (blue) were captured by fluorescence microscopy. Representative images are shown. Scale bar: 100 µm. (**E**) Mean gray value of single-cell fluorescent intensity after PLIN1 labeling was quantified using ImageJ. Data are shown as mean ± SEM. Mann–Whitney U test significance levels: not significant (ns), *p* < 0.0001 (****).

**Figure 3 ijms-23-09218-f003:**
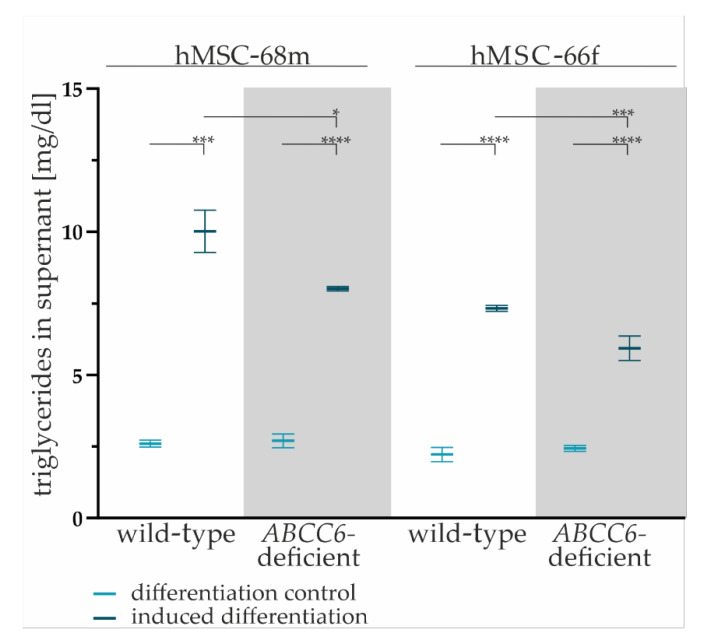
The TG content in cell culture supernatants. Cell culture supernatants were collected from wild-type and *ABCC6*-deficient (grey highlighted) hMSCs after final cultivation in maintenance medium at day 19 of adipogenic differentiation. The content was reduced by basal TG content in maintenance medium. Data are shown as mean ± SEM. Mann–Whitney U test significance levels: not significant (ns), *p* < 0.05 (*), *p* < 0.001 (***), *p* < 0.0001 (****).

**Figure 4 ijms-23-09218-f004:**
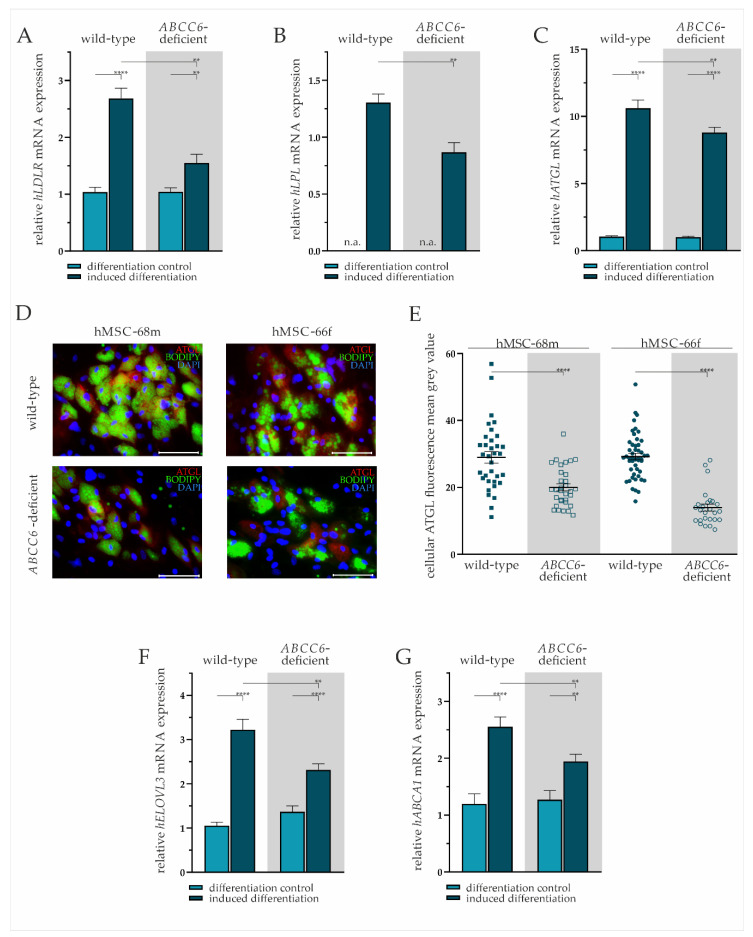
Gene expression analysis and immunofluorescent staining in wild-type and *ABCC6*-deficient adipocytes. (**A**) *LDLR*, (**B**) *LPL*, (**C**) *ATGL*, (**F**) *ELOVL3,* and (**G**) *ABCA1* mRNA expression in undifferentiated hMSCs (differentiation control) and adipocytes (induced differentiation) was measured via RT-qPCR. Wild-type and *ABCC6*-deficient hMSCs (*n* = 2, grey highlighted) were cultured for differentiation over a period of 19 days. (**D**) Immunofluorescent staining for ATGL (red), lipid staining via Bodipy (green), and nuclear counterstaining with DAPI (blue) were captured via fluorescence microscopy. Representative images are shown. Scale bar: 100 µm. (**E**) Mean gray value of single-cell fluorescent intensity after ATGL labeling was quantified using ImageJ. Data are shown as mean ± SEM. Mann–Whitney U test significance levels: *p* < 0.01 (**), *p* < 0.0001 (****). Relative mRNA expressional level below detection threshold is indicated as not available (n.a.).

**Table 1 ijms-23-09218-t001:** Characteristics of hMSCs used in this study.

Nomenclature	Origin	Age	Sex	Lot	Surface Markers *
hMSCS-68m	Human bone marrow/femoral head	68	male	467Z023.5	CD73^+^, CD90^+^, CD105^+^, CD14^−^, CD34^−^, CD45^−^, CD19^−^
hMSCS-66f	Human bone marrow/femoral head	66	female	451Z012.3	CD73^+^, CD90^+^, CD105^+^, CD14^−^, CD34^−^, CD45^−^, CD19^−^

* ≥90% positive for positive markers, ≤10% positive for negative markers.

**Table 2 ijms-23-09218-t002:** Primer sequences, annealing temperatures (TA), efficiency, and resulting product sizes used for qRT-PCR.

Gene	5′-3′ Primer Sequences	TA (°C)	Efficiency	Product Size (bp)
*hABCA1*	ATCCCCAGCACAGCCTAT TCTCCCCAAACCTTTCCA	59	1.88	212
*hABCC6*	CCTGCTGATGTACGCCTT ACGCGAGCATTGTTCTGA	60	1.92	267
*hATGL*	CAGCGGTTTCATCCCCGTGT GCACATCTCTCGCAGCACCA	61	2.00	283
*hELOVL3*	CAGCTTTGCAAGTCCGCGTTA GAGACGGAACAGAGCCGGGA	66	1.75	136
*hHMBS*	CTGCCAGAGAAGAGTGTG AGCTGTTGCCAGGATGAT	63	1.92	165
*hLDLR*	CGACTGCAAGGACAAATCTG AGTCATATTCCCGGTCACAC	61	2.00	122
*hLEP*	GGGAACCCTGTGCGGATTCTT GGAGGAGACTGACTGCGTG	66	1.81	408
*hLPL*	CCCTGCTCGTGCTGACTCTG CGCGGACACTGGGTAATGCT	61	2.00	362
*hPLIN1*	CGGTCAGCCGGAGTGAGTG TACTGGAGGGCGGGGATCTT	66	2.00	417
*hPPARg*	ACCAAAGTGCAATCAAAGTGGA ATGAGGGAGTTGGAAGGCTCT	66	1.96	100
*hRPL13*	CGGAAGGTGGTGGTCGTA CTCGGGAAGGGTTGGTGT	63	1.87	115
*hSDHA*	AACTCGCTCTTGGACCTG GAGTCGCAGTTCCGATGT	63	1.93	177

**Table 3 ijms-23-09218-t003:** Primer sequences, annealing temperature (TA), and product sizes for HotStar DNA polymerase kit amplification.

Gene	5′-3′ Primer Sequences	TA (°C)	Product Size (bp)
*ABCC6*	CTGTTCTCCGGGCATCAGAG GATGGACGGGGTGGTAGGAT	59	554
*GRIN1*	GCGCCGCTAACCATAAACAA GAATCTCCTGCGGAGGGACG	56	213
*HMGN5*	AGCAGATGCTTGTGCCAGTT CCCCACCCAAGGGGTTTAC	56	218
*PIEZO2*	GCCCTGGAACTGGTGGTCTT AAAGGCTTCCCACTCTCAACT	56	144

## Data Availability

The original raw data and materials presented in the study will be made available upon request. Further inquiries can be directed to the corresponding author.

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
