# Peer review of "Targeting ABCC6 in Mesenchymal Stem Cells: Impairment of Mature Adipocyte Lipid Homeostasis"

_ijms, 2022, doi:10.3390/ijms23169218_

Round 1
Reviewer 1 Report
Pseudoxanthoma elasticum (PXE) is a rare disorder of ectopic calcification --- patients develop calcification in the elastic fibers in their skin, eyes, and vessels leading to characteristic skin lesions and laxity, central vision loss, and peripheral vascular disease. While it has long been established that biallelic mutations in ABCC6 cause PXE, the pathological mechanisms driving disease remain elusive. In their paper, Plumers et al. focused on the effect of ABCC6 deficiency on adipocytes and lipid homeostasis. They used CRISPR-cas9 to knock-out ABCC6 in hMCSs and then differentiated these cells into adipocytes to look for in vitro differences in lipid homeostasis. They found that ABCC6-deficient cells had decreased PLIN1 expression compared to controls, however both cell lines essentially progressed through adipogenic differentiation similarly. Interestingly, despite reduced gene expression, ABCC6-deficient cells had increased PLIN1 staining. Notably, ABCC6-deficient cells had less secreted triglycerides in the supernatant compared to controls suggesting intracellular lipid accumulation and impaired lipolysis. In addition, the ABCC6-deficient cells had different gene expression of key players in adipogenesis compared to controls. Overall, Plumers et al. demonstrate that ABCC6-deficient cells have dysregulated lipid metabolism in vitro. While most of the evidence is correlative without functional confirmation, this paper could be considered for publication, if the following comments are comprehensively and adequately addressed.
Major comments
1. The CRISPR-cas9 generated ABCC6 knock-out hMSCs were not a pure/clonal population. There were both wildtype and heterozygous cells mixed in to the ABCC6-deficient cell line which could significantly confound results. This limitation was stated in the Results, but should also be included in the Discussion.
2. A majority of the results are comparing gene expression profiles in control/wildtype v. ABCC6-deficient-differentiated adipocytes. For some of the gene expression differences (such as ATGL), IF was done to confirm the mRNA findings. However, for a number of other differences, there was no confirmation done at the protein or functional level. The authors claim that gene expression patterns informed lipid uptake, intracellular fatty acid trafficking/processing, and cholesterol efflux, however their results are really just correlative. It might be helpful to further investigate some of the gene expression changes at the protein and functional level to substantiate their claims. (Even within their own data, PLIN1 gene expression is decreased, but protein levels as demonstrated by IF were increased.) If these conformational studies are not possible, then tempering their conclusions and emphasizing the correlative nature of the data would be advisable.
Minor comments
There are many typos and spelling errors, to point out a few:
1. There is a typo on page 2, line 50. “aFbsored”
2. There is a spelling error on page 2, line 52. “assembked”
3. There is a diction error on page 4, line 145. “either thymine or cytosine four, respectively, three base pairs upstream”
4. There is a typo on page 4, line 150. “thr”
5. There is a typo on page 4, line 154. “therelative”
6. There is a spelling error on page 14, line 463. “allel”
Author Response
Point 1: The CRISPR-cas9 generated ABCC6 knock-out hMSCs were not a pure/clonal population. There were both wildtype and heterozygous cells mixed in to the ABCC6-deficient cell line which could significantly confound results. This limitation was stated in the Results, but should also be included in the Discussion.
Response 1: We thank the reviewer very much for this advice and agree that for practical reasons it was not possible to work with a pure/clonal population. Nevertheless, we assume that the influence of the wild-type proportion on the phenotype of the whole population is limited and can be reliably assessed with the mutation efficiency and the quantification of ABCC6 mRNA expression.
As requested, we have taken up this consideration and discussed this limitation. It can be found in line 277-284 on page 9.
Point 2: A majority of the results are comparing gene expression profiles in control/wildtype v. ABCC6-deficient-differentiated adipocytes. For some of the gene expression differences (such as ATGL), IF was done to confirm the mRNA findings. However, for a number of other differences, there was no confirmation done at the protein or functional level. The authors claim that gene expression patterns informed lipid uptake, intracellular fatty acid trafficking/processing, and cholesterol efflux, however their results are really just correlative. It might be helpful to further investigate some of the gene expression changes at the protein and functional level to substantiate their claims. (Even within their own data, PLIN1 gene expression is decreased, but protein levels as demonstrated by IF were increased.) If these conformational studies are not possible, then tempering their conclusions and emphasizing the correlative nature of the data would be advisable.
Response 2: We would like to thank you for pointing out the limitation of our study regarding the correlation of the influence of ABCC6 deficiency on lipid metabolism.
Indeed, the confirmation of the results we could see at the gene expression level was not trivial at the protein level. In addition to immunofluorescence-based detection of perilipin, we tried to use the same antibody to quantify it by Western blot. However, we found that the band intensity not only of perilipin but also of the loading control (GAPDH) varied considerably within biological replicates. Protein extraction from fatty tissue for western blot is notoriously difficult and is affected by interferences from lipids (Marin et al., RELi protocol: Optimization for protein extraction from white, brown and beige adipose tissues, MethodsX, Volume 6, 2019, Pages 918-928https://doi.org/10.1016/j.mex.2019.04.010). A similar phenomenon can only be suspected here and would require time-consuming establishment for protein extraction in this particular context.
For the protein detection by immunofluorescence, we decided to limit to the ATGL, since we were able to successfully establish an antibody for this which could be reliably quantified. Meanwhile, we had performed an IF-based labelling of the LDLR, too, which did not work well due to potential unspecific binding and inconsistant distribution within single cells and along a population.
Even with regard to these difficulties, we acknowledge your point that our effects are just correlative. Since this study had less of a claim to elucidate the underlying pathomechanism and to fathom the regulatory network in the background, but is primarily dedicated to the discovery that ABCC6 might play a relevance in adipocyte metabolism, we refrain from further consolidating the gene expression data with protein-based evidence, not least for time- and cost- costuming cultivation, differentiation and establishment.
Last but not least, we have therefore included the above-mentioned points in our discussion to the best of our knowledge and focused more on the correlative nature of our data. Please find the corresponding changes on page 9, line 298-301, page 10 line 350-351, page 11 line 361+368+369, page 11 line 372-374.
Reviewer 2 Report
The manuscript authored by Plümers et al., reports the role of ABCC6 deficiency during adipogenic differentiation of human bone marrow derived stem cells (hMSCs). The manuscript is potentially interesting indicating a possible involvement of ABCC6 in lipid homeostasis. Abstract is clear and concise; titles are informative and reflects the description. Experiments are well designed, executed and the conclusions drawn are supported by the results. I recommend and believe that the manuscript can be accepted for publication in International journal of molecular sciences.
The strength of the present study is that this is the first study indicating that ABCC6 deficiency results in impaired lipid trafficking and suggesting it is as relevant target in pseudoxanthoma elasticum (PXE).
Although authors did not use any in vivo approach to support their claim, but they comprehensively used solid in vitro system to strongly support their argument. To study the importance of ABCC6 authors used CRISPR-Cas9 genome editing system to ablate its expression.
Using ABCC6-deficient hMSCs and wildtype control they give several important insights into the adipogenic differentiation particularly its importance in lipid homeostasis by analyzing triglycerides content and lipid uptake and trafficking markers which was significantly down regulated in ABCC6-deficient hMSCs during differentiation.
Although it would have been more interesting and informative if authors could perform RNAseq analysis particularly after day 19 after differentiation (as ABCC6 expression is hugely induced at this time point) in wildtype and ABCC6-deficient hMSCs. The differentially expressed genes obtained by RNAseq and subsequent pathway analysis would have given much clearer picture about lipid synthesis, uptake, transport etc. Presently, authors have used most of essential markers which could be a limitation of the study.
Author Response
Point 1: Although it would have been more interesting and informative if authors could perform RNAseq analysis particularly after day 19 after differentiation (as ABCC6 expression is hugely induced at this time point) in wildtype and ABCC6-deficient hMSCs. The differentially expressed genes obtained by RNAseq and subsequent pathway analysis would have given much clearer picture about lipid synthesis, uptake, transport etc. Presently, authors have used most of essential markers which could be a limitation of the study.
Response 1: We would like to thank you for your constructive comments and agree that the analysis of essential markers of lipid trafficking is a limit of this study.
Since this study had less of a claim to elucidate the underlying pathomechanism and to fathom the regulatory network in the background, but is primarily dedicated to the discovery that ABCC6 might play a relevance in adipocyte metabolism, we have decided to limit our study to these markers.
Certainly, one of the next necessary steps would be to further characterize this influence in terms of causal relationships. RNA sequencing, as suggested by you, would definitely be helpful, but in our case it would have been beyond the scope of the study.
We have therefore included your point as an outlook in the discussion on page 11 line 372-374.